# Comparative Study of Sperm Selection Techniques for Pregnancy Rates in an Unselected IVF–ICSI Population

**DOI:** 10.3390/jpm13040619

**Published:** 2023-03-31

**Authors:** Ioana Zaha, Petronela Naghi, Liana Stefan, Cosmina Bunescu, Mihaela Radu, Mariana Eugenia Muresan, Mircea Sandor, Liliana Sachelarie, Anca Huniadi

**Affiliations:** 1Department of Medical Dicipline, Calla—Infertility Diagnostic and Treatment Center, Constantin A. Rosetti Street, 410103 Oradea, Romania; drzahaioana@gmail.com (I.Z.); petronelanaghi@gmail.com (P.N.); lianaantal@gmail.com (L.S.); mirelamihaela@yahoo.co.uk (M.R.); ancahuniadi@gmail.com (A.H.); 2Faculty of Medicine and Pharmacy, University of Oradea, 1st December Square 10, 410073 Oradea, Romania; 3Department of Prelinical Discipline, Apollonia University, 700511 Iasi, Romania; 4Department of Medical Dicipline, Pelican Clinical Hospital, Corneliu Coposu Street 2, 410450 Oradea, Romania

**Keywords:** spermatozoa, fertilization, embryo, pregnancy

## Abstract

(1) Background: Semen analysis is a poor predictor of the fertilization potential of spermatozoa and a male factor may contribute to poor outcomes of the IVF procedure, despite a normal semen analysis. The microfluidic sperm selection (ZyMot-ICSI) is based on the selection of the spermatozoa with the lowest DNA fragmentation rate, but studies do not prove better clinical outcomes after this method. (2) Methods: We conducted a retrospective trial comparing 119 couples that were allocated to the classic gradient centrifugation sperm method (control group), and 120 couples that were allocated with the microfluidic technique being used (study group) at our university-level clinic, to go through IVF. (3) Results: The statistical analysis showed that there is no significant difference between the fertilization rate (study vs. control *p* = 0.87), but regarding blastocyst rate (study vs. control *p* = 0.046) and clinical pregnancy (*p* = 0.049), there is quite a significant statistical difference. Microfluidic preparation of spermatozoa seems to improve the results and it may be utilized more broadly for ICSI, and could also improve the workflow in standard IVF, decrease intervention by laboratory personnel and provide more consistent incubation conditions. (4) Conclusions: The patients that had the sperm preparation for ICSI with the microfluidic sperm selection had slightly better results compared with the gradient centrifugation selection.

## 1. Introduction

If infertility is due to low sperm quality, ART treatments (assisted reproductive technology), such as in vitro fertilization (IVF) or intracytoplasmic sperm injection (ICSI), seem to be a viable option for achieving fertilization [1,2,3]. By studying and understanding ART techniques, research has been carried out in the development of new technologies that support their success, but recent data prove that there are still significant deficiencies to reach satisfactory pregnancy rates [2,3,4].

Male factor infertility is a common but little understood aspect. Fertility is not strictly determined by conventional sperm analysis, this being only one of the factors. Thus, determining sperm quality can be one of the aspects of sperm investigation that can have an impact on embryonic development [3]. A complete understanding of male infertility and the success of IVF treatment is given primarily by sperm analysis and using microfluidics could eliminate the mechanical stress induced on sperm during preparation, such as the centrifugation step. The characteristics of the forward movement of spermatozoa underly the use of microfluidic devices.

The preparation of spermatozoa during the in vitro fertilization process is an important part, and classically requires the gradient centrifugation sperm method, a chemical process which increases the levels of oxygen radicals [5]. In the phase of an in vitro fertilization (IVF) cycle, centrifugation and upward swimming are used to collect the highly motile sperm fraction and remove impurities. An important clue in obtaining a higher pregnancy rate is related to sperm selection, especially in male infertility. Sperm sorting, recently performed based on a microfluidic chip, appeared as an alternative tool to the historical sperm preparation methods [1,2,3,4]. The microfluidic technique can rapidly isolate poor sperm samples with high motility, high DNA integrity and low morphological abnormalities [3,4,5]. Microfluidic sperm processing is a sperm separation procedure in which highly motile spermatozoa are isolated from an unprocessed sample and sperm concentrations are constantly reduced, reflecting the highly selective nature of the device [4]. Microarray sperm sorting can more closely replicate in vivo physiological conditions to improve sperm selection and increase the possibility of a successful ICSI outcome [6]. A microfluidic sorter must not alter sperm specifications such as motility, morphology, DNA integrity and the acrosome. A new technique, called the microfluidic technique, promises a better selection of spermatozoa with a lower DNA fragmentation index [5,6]. There are few data and studies that compare the classic sperm selection (the gradient centrifugation) versus the microfluidic technique, and further studies are necessary to establish if there is a significant difference in fertilization, embryo quality, clinical pregnancy rate and live birth between the two techniques in sperm preparation during ICSI (intracytoplasmic sperm injection) in IVF (in vitro fertilization) [7,8,9,10]. The gradient centrifugation sperm method uses chemical and centrifuge stages in preparing the spermatozoa, and it tries to mimic the normal genital tract and the passage of sperm during live fecundation. In contrast, the new technique selects spermatozoa with a higher DNA integrity. Classical sperm selection does not take into consideration DNA fragmentation [11].

ICSI (intracytoplasmic sperm injection) is a largely used technique that has established lower miscarriage rates and higher fertilization rates since 1992. In theory, using the microfluidic technique with low-oxygen radicals and using ICSI for sperm selection could result in higher success rates overall [12].

The purpose of the study is to compare two groups: the control group that includes patients that have the ICSI procedure and the sperm selection is based on the gradient centrifugation sperm method, and the study group that includes patients that have the ICSI procedure performed and the selection made was with the microfluidic technique, all being applied to a general IVF population that has not been selected [13,14]. The microfluidic technique has the advantage of selecting most naturally the spermatozoa with the lowest DNA fragmentation [15,16,17,18]. Having a new and improved selection method gives a better chance of using sperm with less DNA fragmentation and improving the fertilization rate [19]. To have a greater effect on sperm preparation during ICSI, one of the critical parameters is the post-preparation sperm count [20,21,22,23]. 

The study’s primary outcome is to establish if there are noticeable differences between groups in the fertilization rate of oocytes, and the number and embryo quality. The secondary outcome is the clinical pregnancy rate.

## 2. Materials and Methods

### 2.1. Population

This is a 12-month retrospective study in a heterogeneous population, comparing the control group in which the gradient centrifugation sperm method was used for sperm selection for ICSI during IVF, and the study group in which ZyMot (microfluidic sperm selection device) was used. In this retrospective study, we compared 120 couples that underwent an in vitro procedure at our clinic, whereby the classic gradient centrifugation sperm method was used (control group) with 120 couples that underwent an IVF procedure and the microfluidic technique was used for other couples (study group). All couples had a clear indication to go through IVF at our university-level clinic, and no type of pathology was excluded, making both groups non-bias. We conducted a retrospective comparative study for sperm preparation using a microfluidic sperm separation device, compared with density gradient centrifugation in a subject undergoing an IVF procedure at a single university-level IVF clinic in Oradea, Romania. 

The inclusion criteria were: women between 21 and 40 years of age, males partners aged between 21 and 45 years, diagnosed with infertility requiring IVF and willing to sign the participation agreement. 

Exclusion criteria were: females over 41 years of age, and a male partner with severe oligoasthenospermia and cancer diagnosis. Enrollment lasted from 1 January 2022 until 31 December 2022. We chose the age of 41 as the upper limit of inclusion in the study because up to this age, we performed ovarian stimulation and obtained embryos with each woman’s genetic material (own oocytes). After this age, we offer them the possibility of donated oocytes. However, we know that the rate of aneuploidy after 38 years is higher, and we offer the possibility of genetic testing of the embryos. During this timeframe, 280 patients were eligible to participate, and 239 signed the consent form for participation. They were randomized in a 1:1 ratio to standard sperm processing or microfluidic sperm preparation, resulting in 119 subjects undergoing sperm processing with density gradient (control group) and 120 microfluidic sperm preparation, ZyMot (study group). All the patients required IVF and the etiology of infertility was various, and they all had ICSI cycles. 

The present study observed the ethic conditions established by the Helsinki Declaration, being approved by the local ethic committee of Calla, Infertility Diagnostic and Treatment Center of Oradea no. 638/25.11.2021, and informed consent was obtained before the inclusion of the participants in the study.

### 2.2. Methods

In this retrospective study, we compared 119 couples that underwent an in vitro procedure in our clinic, and the classic gradient centrifugation sperm method was used (control group) with 120 couples that underwent an IVF procedure, and the microfluidic technique was used for the rest (study group). All couples had had a clear indication to go through IVF at our university-level clinic, and no type of pathology was excluded, making both groups non-bias. The patients in both groups had a stimulation protocol and the HCG (human chorionic gonadotropin) trigger followed by the ovarian puncture at 34–36 h. After the oocyte recovery and then the examination, the partner was asked to give a semen sample by masturbation in a sterile plastic container, for the ICSI to be performed. 

In the control group after liquefication at 37 °C for 10–30 min, a first swim-up in G-IVF plus was performed in an incubator under 6% CO_2_, and after that submitted for the second migration under MSU (micro swim-up). A dish is prepared for oocytes in an H pattern, and after that, IVF (Vitrolife) is prepared for semen. A small amount of semen was placed on the side of the H and incubated for 3 min, so the sperm migrates on the outside of the H. The pipeline was used to immobilize the sperm by the tail and inject it into the oocyte. 

Fertilization was assessed at 18 h by the possession of two pronuclei. At day 5, the embryos were graded according to the Gardner criteria. According to the treatment plan, a fresh or a frozen embryo transfer was planned. If the fresh embryo transfer was to happen, luteal support was administered with progesterone on the evening of oocyte retrieval, and continued at least until the day of the HCG test. The frozen embryo transfers were performed in a natural cycle or medicated. The purpose of the natural cycle was to monitor the natural ovulation by the LH surge, with the estradiol level over 200 and progesterone around the value of 1 when the progesterone commenced. In the frozen embryo transfer preparation, a daily dosage of 6 mg of estradiol tablets was required beginning from the second day of a menstrual cycle, with no ovarian cysts observed by ultrasound. After 15 days, if the endometrium thickness reached a value over 7 mm, we started the luteal phase support with 1200 mg of progesterone intravaginally, and on the fifth day, the thawed embryo transfer was performed. Clinical pregnancy was defined by ultrasound confirmation of the gestational sac, with an embryo and fetal heartbeat by transvaginal ultrasound.

During this timeframe, 280 patients were eligible to participate and 239 signed the consent forms for participation. They were randomized in a 1:1 ratio for standard sperm processing or microfluidic sperm preparation, resulting in 119 subjects subject to sperm processing with density gradient (control group), and 120 to microfluidic sperm preparation, ZyMot (study group). All the patients required IVF and the etiology of infertility was various, and they all had ICSI cycles. 

The ovarian stimulation for IVF was performed using a GnRH antagonist protocol. The dose was determined by the patient’s age, BMI and ovarian reserve. The GnRH antagonist (0.25 mg Cetrotide, EMD-Sereno Inc., Rockland, MA, USA) was administrated from day 5 or 6 of stimulation, depending on the hormonal status and ultrasound findings (estradiol over 300 and a leading follicle of over 14 mm), to prevent premature ovulation. 

Induction of ovulation was performed when the leading follicle was at 18mm in diameter and the hormonal levels had a sufficient estradiol value. Oocyte retrieval was performed after 34–36 h from the ovulation induction via transvaginal punction under ultrasound guidance.

### 2.3. Laboratory Procedures

In the study, group sperm were processed with a single used ZyMot ICSI Sperm Separation Device. It contains five individual channels, of 2 µL capacity per channel. The device is loaded with the Sperm Washing Medium before use. A total of 10 µL of semen was used per patient, after incubation at 37 °C for 30 min. The dimensions of microchannels allowed motile sperm to be selected. If insufficient spermatozoa were obtained, the unprocessed sample remains were processed by the traditional method. Having the semen prepared, the cumulus was stripped at 3 h post-retrieval and all mature oocytes went through ICSI. Sperm samples were processed using 40% and 80% vitrolife grade sperm gradient layers. The density gradient solution constituents contained a colloidal silica particle suspension, adjusted with covalently bound hydrophilic silane provided in HEPES. A sperm wash medium (G-IVFTM) was used to clear and resuspend the final pellet. The gradient medium, sperm wash medium and sperm samples were stored in an incubator at 37 °C and 6% CO_2_ atmosphere for 20 min, for equilibration before the procedure. Briefly, 2 mL of bottom-layer gradient (80%) was transferred to a conical bottom tube. A second layer of 2 mL of the upper layer (40%) was then slowly placed over the lower layer. A distinct line was observed, separating the two layers. An appropriate volume of liquefied semen was placed lightly over the top layer. The prepared tube was then centrifuged at 1100 rpm for 15 min. The supernatant seminal plasma was discarded and each layer of the gradient was collected separately and washed with 5 mL of the Vitrolife G-IVF Medium at 1100 rpm for 10 min. The sperm pellet obtained after each centrifugation was stored separately. For ICSI, the sperm sample from the bottom (second) layer was used.

#### Statistical Analysis

The data were statistically analyzed using SPSS 26 (IBM SPSS Statistics for Windows, Version 26.0. IBM Corp., Armonk, New York, NY, USA). Statistical significance was considered at the standard 5% critical level (0.05). To compare the numerical variables between the 2 groups, we used the non-parametric Mann–Whitney test, because the values of distributions did not follow the normal distribution law. Variables were expressed as numbers and percentages, while continuous variables were expressed as the mean and standard deviation (min–max, where applicable). Statistical significance for all data analyses was set at *p* < 0.05. Differences in fertilization, blastocyst utilization, oocytes and pregnancy outcomes between the study group and the control group were analyzed using a paired *t*-test.

## 3. Results

### 3.1. Predictors

Baseline characteristics including female age, infertility diagnostic, number of oocytes, blastocyst, pregnancy rate and fertility rate were collected. Male characteristics were: age and sperm concentration in the unprocessed sample used for treatment were recorded. The average age of males and females in the control group were 35.62 ± 4.495 and 34.08 ± 4.923 in the study group, respectively (Table 1).

The separate stages of the research are given in Figure 1.

### 3.2. Comparisons between Study and Reference Groups

The number of collected oocytes collected was similar in the two groups (*p* = 0.1282) (Table 2). The similar number of oocytes collected in the two groups has shown the heterogenicity of the groups and the non-bias inclusion in the two groups of female IVF indication. Additionally, the number of oocytes retrieved is similar by using similar stimulation protocols and does not change the outcomes of the patients, as well as the similar AMH and AFC count in the two groups. This difference is not considered to be statistically significant.

No significant differences were observed in the total number of blastocysts (*p* = 0.0462) (Table 2). The *p*-values regarding the fertilized oocytes are not considered to be statistically significant by conventional criteria (Table 2).

Unlike the other parameters, the post-processing concentration regarding the difference between the two groups appears to be very statistically significant (*p* = 0.002) (Table 2). The microfluidic sperm selection technique has the advantage of having a way to use the spermatozoa with low oxidative stress and not only select the morphologically best spermatozoa, but also the ones with a low DNA fragmentation index. Having that criterion of selection, the techniques deliver a high concentration for use in IVF–ICSI.

As depicted in Table 2, no significant differences were observed in the fertilization rates and by conventional criteria, this difference between the groups is not considered to be statistically significant. Table 2 shows a *p*-value of 0.8778 and concludes the fact that both groups have a similar fertilization rate (80.13% in study group vs. 79.79% in the control group). The *p*-value for the pregnancy rate regarding the two groups is considered to be quite statistically significant (*p* < 0.05), demonstrating the higher pregnancy rate in the study group (Table 2). This could also be influenced by higher-quality embryos, but further research is needed.

Between the two groups, we had a slightly better number of blastocysts, but the number of fertilized oocytes and pregnancy rates were almost the same, with no significant difference between the groups. The biggest plus in the new microfluidic technique usage in ICSI during IVF is that it can lead to an easier selection of spermatozoa that have better quality, lower DNA fragmentation, and could lead to a higher number of embryos.

The *p*-value for the motility A + B (%) regarding the two groups is considered to be quite statistically significant (*p* < 0.05), indicating that the treatment of infertility can be successful in the study group (Table 2). In particular, sperm motility seems to be the best male parameter to predict fertilization rates in IVF. This could suggest that for the male partner, an important improvement of sperm motility could be essential in the IVF outcome.

## 4. Discussion

The interest in a new sperm selection technique will give a better outcome of IVF in the future. In vitro fertilization (IVF) has become an interesting scientific achievement of the 20th century with a major impact on human lives [6]. IVF involves a series of fairly complicated procedures that are appropriate to treat infertility and genetic problems that help with childbirth. This type of treatment also involves a deep emotional and physical aspect for women and their partners. Research shows that most couples who get involved in such a program are well-adapted from a psychological point of view [21]. We strongly believe that there is a plus in every step of IVF-related discoveries, and even if ZyMot helps embryologists select better sperm, despite the no known difference in pregnancy rates as it is at this point, this process gives us hope for higher quality blastocysts.

Sperm selection may be an important factor, especially in infertility cases where the male factor is present; however, the methodologies developed to date have not proven to be useful for their routine application in clinical practice and seem to be effective only in specific cases of male infertility [22].

One major limitation would be that there is no sufficient data gathered at this point about the efficiency of the microfluidic technique during IVF, and that studies on a larger population and multicentric studies for this information should be performed. Another limitation is in the real assessment of the pregnancy rate. The pregnancy rate itself is due to various factors and can be influenced by other parameters besides the usage of one or another technique in sperm selection. The high sperm DNA fragmentation is known to have a negative impact on IVF outcome, still, testing for it is not recommended in every case [21,22].

The criteria for male patients in this study were being aged between 21 and 45 years, with an infertility diagnosis as a couple. As it is already known, normal sperm count is a low predictor of fertilization rate. That is why even in patients with normal sperm count, the usage of ICSI was preferred for the maximization of the fertilization rate, having a higher fertilization rate than regular IVF. The European Society of Human Reproduction and Embryology (ESHRE) explains that “ICSI continues to be the preferential fertilization method around the world, even in the absence of the male factor. Current guidelines still cannot justify this tendency” (ESHRE, 2020) [23].

This study aims to improve knowledge of clinical outcomes and treatment in IVF with a heterogeneous population. This could generalize the results in their use by doctors. As a single-center design, a multicenter study may provide greater generalizability and applicability for the microfluidic sperm selection process and may have access to a larger study group. For future studies, the pregnancy rate could be the primary outcome. Even if microfluidic processing did not drastically improve the results, it could improve the workflow and provide more consistent incubation conditions. It is an easy procedure, with high applicability and repeatability that mimics the natural selection of sperm traveling through the cervix to the fallopian tubes. [24,25,26,27].

The fertilization rate was similar between the groups, with a slightly higher pregnancy rate in the study group. The microfluidic sperm selection had a higher number of high-grade embryos that were used in frozen embryo-transfer cycles, and produced a higher cumulative pregnancy rate. This was an inconsistent find in other studies being reported as well, with a higher pregnancy rate resulting in some research studies [28].

The microfluidic sperm sorting technique represents an important stage in major infertility studies, and treatments with spermatozoa which are viable, mobile and morphologically appropriate must be separated from defected sperm for fertilization [29]. Recently, many significant advances have been made in the potential utility of microfluidics in the isolation, manipulation, analysis and cryopreservation of gametes and embryos [30].

The proposed methods each have their advantages and disadvantages, and are still under study. The method used should be based on the characteristics of each ICSI case. Thus, the microfluidic technique did not change the outcome of IVF, and larger studies need to be done in selected populations with abnormal sperm parameters and/or high DNA fragmentation in order to establish whether the microfluidic technique helps and improves the IVF outcome.

## 5. Conclusions

At this point, ZyMot is an add-on in IVF that can be used, and seems to slightly improve the number of the good-quality blastocyst and pregnancy rates, but further studies are needed to confirm this. Additionally, more research is needed to fully understand the potential risks and benefits of microfluidic sperm preparation and its impact on embryo quality and pregnancy outcomes.

## Figures and Tables

**Figure 1 jpm-13-00619-f001:**
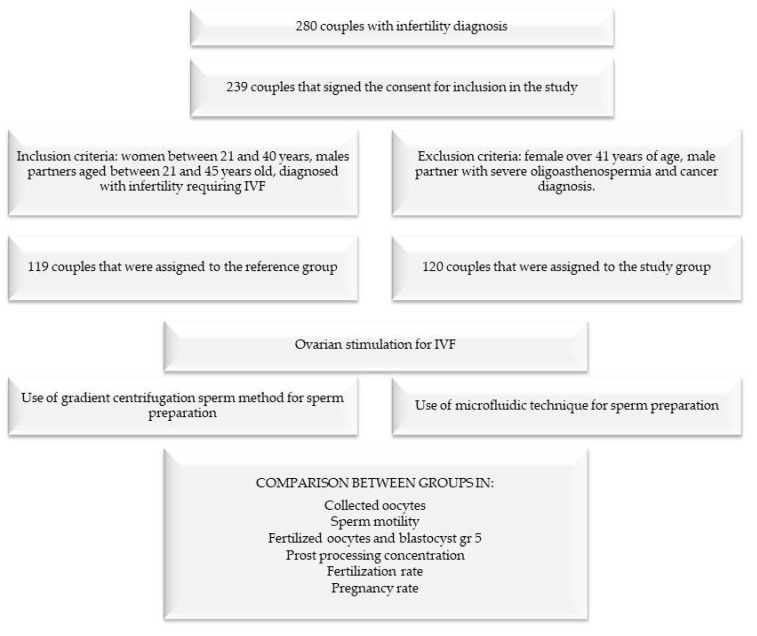
Workflow diagram.

**Table 1 jpm-13-00619-t001:** Baseline statistical characteristics for control and study groups.

Age	N	Mean	Std. Deviation
Control group	119	35.62	4.495
Study group	120	34.08	4.923
Initial concentration (millions/mL)			
Control group	119	39.679	35.51766
Study group	120	40.231	23.80462
Post-processing concentration			
Control group	119	24.462	27.0717
Study group	120	16.312	11.4606
Oocytes collected			
Control group	119	9.66	6.626
Study group	120	10.98	6.819
Motility A + B (%)			
Control group	119	46.84	23.76532
Study group	120	57.00	17.39545

**Table 2 jpm-13-00619-t002:** Comparations between control group and study group.

Oocytes Collected	Study Group	Control Group	t	df	*p*-Value
Mean	10.98	9.66	1.5265	237	0.1282
SD	6.82	6.63			
SEM	0.62	0.61			
Total number of blastocysts					
Mean	2.66	1.63	0.3028	109	0.0462
SD	1.71	0.15			
SEM	0.16	2.28			
Post-processing concentration					
Mean	16.312	24.877	3.1268	231	0.002
SD	11.461	27.496			
SEM	1.051	2.575			
Fertilization rate					
Mean	80.13	79.79	0.1539	238	0.8778
SD	17.01	17.38			
SEM	1.55	1.59			
Cumulative pregnancy rate					
Mean	51.0415	40.2080	1.8916	238	0.049
SD	45.8039	42.8701			
SEM	4.1813	3.9135			
Motility A + B (%)					
Mean	99.75	35.00	30.0985	120	0.001
SD	2.74	14.14			
SEM	0.25	10.00			

## Data Availability

Not available.

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
