# Peer review of "Comparative Study of Sperm Selection Techniques for Pregnancy Rates in an Unselected IVF–ICSI Population"

_jpm, 2023, doi:10.3390/jpm13040619_

Round 1
Reviewer 1 Report
The manuscript entitled ‘Comparative study of sperm selection techniques for pregnancy rate in an unselected IVF-ICSI population’ is not suitable for publication. It is not correctly written, badly organized and conclusions are useless.
Some modifications to made:
Introduction:
line 48: change ‘improved’ by ‘high’
lines 77-79: please conjugate verbs at the past instead of present as the study is closed. These lines must be written at the end of conclusion, not in the middle. First you introduce the arguments and then the aim of the present study.
Material and methods:
lines 102-110: same thing. The aim of the study belongs to the end of the introduction.
Line 126: why 41 years old? Why not 38 as aneuploidy rate increases in embryos from this maternal age?
Once again, disorganization : you first describe ICSI (lines 143-9) then sperm production and preparation (lines 170-8). No sense!
Methods (2.3) is separated from laboratory procedures (2.4) or statistical analysis (2.5) but 2.4 and 2.5 are methods!
Lines 134-6: 120 couples in both groups but 119 in control group from table 1. This difference is repeated in several lines in table 1.
Draw a scheme to better make understand your work.
Table 1: too many statistical data that make the true data disappear. We don’t care about median or standard error of mean. We need to know if data are statistically different or comparable. If not the 2 groups cannot be compared. Add fertilization rate instead of zygote number.
Tables: too many. They should be all regrouped in only one.
Conclusion: your conclusion is that sperm selection based on DNA fragmentation has no consequences on in vitro results and clinical pregnancy or live birth rates. There is no data about live birth rates in your work. Furthermore, cumulative pregnancy rate is higher for study group.
Sperm DNA fragmentation is known to affect miscarriage probability (Agarwal et al. 2020. World J. Men’s Health). Of course you will find nothing from lab data!
Author Response
The authors acknowledge the useful observations and suggestions of the reviewer’s as concerns the manuscript entitled:
Comparative Study of Sperm Selection Techniques for Pregnancy Rate in an Unselected IVF-ICSI Population
Ioana Zaha 1, Petronela Naghi 1, Liana Stefan 1,2, Cosmina Bunescu 1, Mihaela Radu 1, Mariana Eugenia Muresan 2, Mircea Sandor 2,*, Liliana Sachelarie 3,* and Anca Huniadi 1,2,4
According to the reviewer’s recommendations, all the suggestions were taken into account, as follows:
Introduction:
- line 48: change ‘improved’ by ‘high’
Done
- lines 77-79: please conjugate verbs at the past instead of present as the study is closed. These lines must be written at the end of conclusion, not in the middle. First you introduce the arguments and then the aim of the present study.
Done
Material and methods:
- lines 102-110: same thing. The aim of the study belongs to the end of the introduction.
Done
- Line 126: why 41 years old? Why not 38 as aneuploidy rate increases in embryos from this maternal age?
We chose the age of 41 as the upper limit of inclusion in the study because up to this age, we performed ovarian stimulation and obtained embryos with each woman's genetic material (own oocytes). After this age, we offer them the possibility of donated oocytes. However, we know that the rate of aneuploidy after 38 years is higher and we offer the possibility of genetic testing of the embryos.
- Once again, disorganization : you first describe ICSI (lines 143-9) then sperm production and preparation (lines 170-8). No sense!
Done
We revised the paragraphs.
- Methods (2.3) is separated from laboratory procedures (2.4) or statistical analysis (2.5) but 2.4 and 2.5 are methods!
Done
- Lines 134-6: 120 couples in both groups but 119 in control group from table 1. This difference is repeated in several lines in table 1.
Done
- Draw a scheme to better make understand your work.
Done
|
|
Table 1: too many statistical data that make the true data disappear. We don’t care about median or standard error of mean. We need to know if data are statistically different or comparable. If not the 2 groups cannot be compared. Add fertilization rate instead of zygote number.
Done
Some parameters like the total number of blastocysts, fertilized oocytes, and cumulative pregnancy rate data, are statistically comparable.
Tables: too many. They should be all regrouped in only one.
Done
Conclusion: your conclusion is that sperm selection based on DNA fragmentation has no consequences on in vitro results and clinical pregnancy or live birth rates.
Conclusion: At this point, the ZyMot is an add-on in IVF that can be used and seems to slightly im-prove the number of the good quality blastocyst and pregnancy rate, but further studies are needed. Additionally, more research is needed to fully understand the potential risks and benefits of microfluidic sperm preparation and its impact on embryo quality and pregnancy outcomes.
There is no data about live birth rates in your work. Furthermore, cumulative pregnancy rate is higher for study group.
The birth rate is influenced by numerous maternal factors including endometrial pathology, and uterine or systemic maternal pathology, not only by the quality of the embryo. For this reason, we did not choose the birth rate as the primary outcome, but the pregnancy rate. Indeed, the cumulative pregnancy rate was higher in the study group.
Thank you very much for your review,
Respectfully,
Prof.dr. Liliana Sachelarie

Reviewer 2 Report
The study compared the fertilization, blastocysts formation and pregnancy outcomes after ICSI in 119 couples using the gradient centrifugation sperm selection and 120 couples using microfluidic sperm selection, and concluded that there were no differences between this two groups.
Questions and comments:
1. Suggest using “gradient centrifugation sperm section” rather than “swim-up technique” throughout the text to avoid confusion for general readers since authors used the gradient centrifugation sperm selection method in this study.
2. Noted a plenty of repetitive statement existed in this paper. Suggest deleting lines 80-99 and lines 102-110.
3. In line 134, should it be 119 couples?
4. Make clear description about the gradient centrifugation sperm section method in lines 172-178.
5. Please provide sperm concentration unit in table 1 and 5.
6. Please describe the ICSI selection criteria (indications) for male infertility. It seems that authors perform ICSI even in patients with high sperm concentration shown in table 1. This need to be discussed in discussion, and make sure not misuse the ICSI technology for treating patients with infertility.
Author Response
The authors acknowledge the useful observations and suggestions of the reviewer’s as concerns the manuscript entitled:
Comparative Study of Sperm Selection Techniques for Pregnancy Rate in an Unselected IVF-ICSI Population
Ioana Zaha 1, Petronela Naghi 1, Liana Stefan 1,2, Cosmina Bunescu 1, Mihaela Radu 1, Mariana Eugenia Muresan 2, Mircea Sandor 2,*, Liliana Sachelarie 3,* and Anca Huniadi 1,2,4
According to the reviewer’s recommendations, all the suggestions were taken into account, as follows:
Questions and comments:
- Suggest using “gradient centrifugation sperm section” rather than “swim-up technique” throughout the text to avoid confusion for general readers since authors used the gradient centrifugation sperm selection method in this study.
Done
- Noted a plenty of repetitive statement existed in this paper. Suggest deleting lines 80-99 and lines 102-110.
Done
- In line 134, should it be 119 couples?
Yes. Thank you very much!
- Make clear description about the gradient centrifugation sperm section method in lines 172-178.
Sperm samples were processed using 40% and 80% VITROLIFE GRADE SPERM gradient layers. The constituents of the density gradient solution contained a colloidal suspension of silica particles adjusted with covalently bound hydrophilic silane provided in HEPES. A sperm wash medium (G-IVFTM) was used to clear and resuspend the final pellet. The gradient medium, sperm wash medium, and sperm samples were stored in an incubator at 37°C and 6% CO2 atmosphere for 20 minutes for equilibration before the procedure. Briefly, 2 ml of bottom layer gradient (80%) was transferred to a conical bottom tube. A second layer of 2 ml of the upper layer (40%) was then slowly placed over the lower layer. A distinct line was observed separating the two layers. An appropriate volume of liquefied semen was placed lightly over the top layer. The prepared tube was then centrifuged at 1100 rpm for 15 min. Supernatant seminal plasma was discarded and each layer of the gradient was collected separately and washed with 5 ml of Vitrolife G-IVF Medium at 1100 rpm for 10 min. The sperm pellet obtained after each centrifugation was stored separately.
- Please provide sperm concentration unit in table 1 and 5.
Done
- Please describe the ICSI selection criteria (indications) for male infertility. It seems that authors perform ICSI even in patients with high sperm concentration shown in table 1. This need to be discussed in discussion, and make sure not misuse the ICSI technology for treating patients with infertility.
The criteria for male patients in this study were age between 21 and 45 years old that had an infertility diagnosis as a couple. As it is already known, a normal sperm count it is a low predictor of fertilization rate. That is why even in patients with normal sperm count, the usage of ICSI was preferred for the maximization of the fertilization rate, having a higher fertilization rate than regular IVF. The European Society of Human Reproduction and Embryology (ESHRE) explains that “ICSI continues to be the preferential fertilization method around the world, even in the absence of the male factor. Current guidelines still cannot justify this tendency” (ESHRE, 2020).
Thank you very much for your review,
Respectfully,
Prof.dr. Liliana Sachelarie

Round 2
Reviewer 1 Report
The manuscript entitled ‘Comparative study of sperm selection techniques for pregnancy rate in an unselected IVF-ICSI population’ has improved but remains messy. It still not suitable for publication.
Material and methods:
lines 80-83: As previously said, the aim of the study belongs to the end of the introduction, not of Material and Methods section.
I would rename the section 2.2 as Population, instead of material. Then the section 2.3 would become Material and Methods.
Lines 127-134 belong to laboratory procedures.
Lines 149-154 belong to population description.
Table 1 still be uselessly full of statistical data.
Results:
Figure 1 is OK.
I don’t understand why ovarian hyperstimulation paragraph is written within the Results section!
Make a choice: speak about the number of fertilized oocytes (the scientific appellation is zygote) or fertilization rate (table 2) but not both.
Lines 203-214: revise English. I don’t understand what means ‘quite statistically significant’!
In general, the manuscript construction lacks scientific rigor.
Author Response
The authors acknowledge the useful observations and suggestions of the reviewer’s as concerns the manuscript entitled
Comparative Study of Sperm Selection Techniques for Pregnancy Rate in an Unselected IVF-ICSI Population, by
Ioana Zaha 1, Petronela Naghi 1, Liana Stefan 1,2, Cosmina Bunescu 1, Mihaela Radu 1, Mariana Eugenia Muresan 2, Mircea Sandor 2,*, Liliana Sachelarie 3,* and Anca Huniadi 1,2,4
According to the reviewer’s recommendations, the suggestions were carefully considered, as follows:
- lines 80-83: As previously said, the aim of the study belongs to the end of the introduction, not of Material and Methods section.
Done
- I would rename the section 2.2 as Population, instead of material. Then the section 2.3 would become Material and Methods.
Done
- Lines 127-134 belong to laboratory procedures.
Done
- Lines 149-154 belong to population description.
Done
- Table 1 still be uselessly full of statistical data.
Done
- Results:
Figure 1 is OK.
Thank you!
- I don’t understand why ovarian hyperstimulation paragraph is written within the Results section!
At: Materials and Methods
- Make a choice: speak about the number of fertilized oocytes (the scientific appellation is zygote) or fertilization rate (table 2) but not both.
Done
- Lines 203-214: revise English. I don’t understand what means ‘quite statistically significant’!
Done
Thank you very much for review reports and for the extremely useful observations and suggestions!
Kind regards,
Prof.dr. Liliana Sachelarie
Reviewer 2 Report
In 2.3.1 Laboratory procedures, please clarify the sperm collected from which layer were used for ICSI.
Author Response
The authors acknowledge the useful observations and suggestions of the reviewer’s as concerns the manuscript entitled
Comparative Study of Sperm Selection Techniques for Pregnancy Rate in an Unselected IVF-ICSI Population, by
Ioana Zaha 1, Petronela Naghi 1, Liana Stefan 1,2, Cosmina Bunescu 1, Mihaela Radu 1, Mariana Eugenia Muresan 2, Mircea Sandor 2,*, Liliana Sachelarie 3,* and Anca Huniadi 1,2,4
According to the reviewer’s recommendations, the suggestions were carefully considered, as follows:
- In 2.3.1 Laboratory procedures, please clarify the sperm collected from which layer were used for ICSI.
For ICSI, the sperm sample from the bottom (second) layer was used.
Thank you very much for review reports and for the extremely useful observations and suggestions!
Kind regards,
Prof.dr. Liliana Sachelarie
Round 3
Reviewer 1 Report
There is a repetition in lines 105-100 and lines 147-152.
Please revise English in a scientific optic in Discussion section.
The manuscript should be now ready for publication.
Author Response
The authors acknowledge the useful observations and suggestions of the reviewer’s as concerns the manuscript entitled:
Comparative Study of Sperm Selection Techniques for Pregnancy Rate in an Unselected IVF-ICSI Population
Ioana Zaha 1, Petronela Naghi 1, Liana Stefan 1,2, Cosmina Bunescu 1, Mihaela Radu 1, Mariana Eugenia Muresan 2, Mircea Sandor 2,*, Liliana Sachelarie 3,* and Anca Huniadi 1,2,4
According to the reviewer’s recommendations, all the suggestions were taken into account, as follows:
There is a repetition in lines 105-100 and lines 147-152.
Done
Please revise English in a scientific optic in Discussion section.
Done
The manuscript should be now ready for publication.
Thank you very much for your review,
Respectfully,
Prof.dr. Liliana Sachelarie
